# Inflammatory Biomarkers and Gait Impairment in Older Adults: A Systematic Review

**DOI:** 10.3390/ijms25031368

**Published:** 2024-01-23

**Authors:** Lorenzo Brognara, Oscar Caballero Luna, Francesco Traina, Omar Cauli

**Affiliations:** 1Department of Biomedical and Neuromotor Sciences (DIBINEM), Alma Mater Studiorum University of Bologna, 40123 Bologna, Italy; francesco.traina@ior.it; 2Department of Nursing, University of Valencia, 46010 Valencia, Spain; oscar.caballero@uv.es (O.C.L.); omar.cauli@uv.es (O.C.)

**Keywords:** cytokines, molecular marker, falls, IL-6, TNF-alpha, inflammation, gait speed

## Abstract

Peripheral inflammation and gait speed alterations are common in several neurological disorders and in the aging process, but the association between the two is not well established. The aim of this systematic literary review is to determine whether proinflammatory markers are a positive predictor for gait impairments and their complications, such as falls in older adults, and may represent a risk factor for slow gait speed and its complications. The systematic review was performed in line with the Preferred Report Items for Systematic Review and Meta-Analyses (PRISMA). A protocol for literature searches was structured a priori and designed according to the International Perspective Register of Systemic Review (PROSPERO: CRD42023451108). Peer-reviewed original articles were identified by searching seven electronic databases: Excerpta Medica Database (EMBASE), SciVerse (ScienceDirect), Scopus, PubMed, Medline, Web of Science, and the Cochrane Library. The search strategy was formulated based on a combination of controlled descriptors and/or keywords related to the topic and a manual search was conducted of the reference lists from the initially selected studies to identify other eligible studies. The studies were thoroughly screened using the following inclusion criteria: older adults, spatiotemporal gait characteristics, and proinflammatory markers. A meta-analysis was not performed due to the heterogeneity of the studies, and the results were narratively synthesized. Due to the clinical and methodological heterogeneity, the studies were combined in a narrative synthesis, grouped by the type of biomarkers evaluated. A standardized data extraction form was used to collect the following methodological outcome variables from each of the included studies: author, year, population, age, sample size, spatiotemporal gait parameters such as gait velocity, and proinflammatory markers such as TNF-α, high sensitivity C-reactive (CRP) proteins, and IL-6. We included 21 out of 51 studies in our review, which examined the association between inflammatory biomarkers and gait impairment. This review highlights the role of TNF-α, CRP, and IL-6 in gait impairment. Biomarkers play an important role in the decision-making process, and IL-6 can be an effective biomarker in establishing the diagnosis of slow gait speed. Further longitudinal research is needed to establish the use of molecular biomarkers in monitoring gait impairment.

## 1. Introduction

Gait velocity is a simple screen of functional status in older adults, and predicts major adverse outcomes in older individuals such as falls, dementia, and death [1,2,3,4,5,6]. A gait speed lower than 0.8 m/s is a reliable cut-off for identifying subjects at increased risk of disability, and evidence on the relationship between circulating IL-6 levels and gait speed suggests that higher IL-6 levels may be associated with poorer performance, hospitalization, institutionalization, and death in older adults [7,8,9,10,11]. Falls are common in older adults, but little is known about accidental falls and their relationship to movement disorders and chronic inflammation. It has been estimated that adults older than 60 years of age suffer the most fatal falls, therefore as the prevalence of older fallers is predicted to increase with changes in demography, prevention strategies should prioritize fall-related research and establish effective strategies to reduce risk [12]. According to the World Health Organization, approximately 684,000 fatal falls occur each year, making it the second leading cause of unintentional injury death after road traffic injuries [13]. The increasing number of people who suffer falls is also an economic problem for health services worldwide, with an estimated cost for the EU of 25 billion Euros for treating fall-related injuries [14]. In addition, people who have experienced a fall develop fear of falling, which is a serious issue that negatively impacts their physical and mental health [15]. Movement disorders and chronic inflammation are frequently comorbid and underdiagnosed [7]. Systemic inflammation is closely associated with central neuroinflammation [8]. The accessibility and practicality of using blood samples have led to numerous studies measuring the profile of serum or plasma immune markers in several neurological disorders, and many of these studies have found a significant association between those markers and disease severity. Other spatiotemporal parameters, such as a stride length of 0.64 m, accurately predict major adverse events such as physical disability, falls, institutionalization, and mortality [16]. In addition, C-reactive protein (CRP) has been significantly and negatively associated with the total number of daily strides [17].

Previous studies of older adults have linked gait and mobility problems to high CRP levels, and chronic inflammation may be a factor affecting denervation of muscles and changes in the neuromuscular junction [18,19]. Inflammation also plays a role in the initiation and progression of knee osteoarthritis, and studies have shown that osteoarthritis is associated with high serum levels of inflammatory markers that alter gait mechanism [20,21,22,23,24,25].

Recent developments in drug therapies with biological agents such as anti-tumor necrosis factor alpha (TNF-α), have provided great benefits in terms of reductions in joint inflammation, pain, and improved gait function [26]. Levels of pro-inflammatory cytokines, and in particular, elevated IL-6 and CRP have been identified as independent predictors of impaired mobility, disability, and slow walking speed in older adults [7,19,27,28,29,30,31,32,33,34,35].

There is growing evidence of associations between elevated levels of inflammatory cytokines such as IL-6, TNF-α, or the acute phase CRP and several chronic health conditions or adverse aging outcomes including muscle loss and cognitive impairment (Figure 1) [34,35].

However, there are only a few studies focusing on peripheral inflammatory processes and gait impairment. The aim of this systematic review is therefore to analyze current scientific knowledge and to investigate the reason that proinflammatory markers represent a positive predictor for gait impairment and its complications. According to the results of this study, there will be further opportunities to develop insights into the pathophysiological mechanisms that make proinflammatory markers such a powerful tool for identifying high-risk people with gait impairments. The results of this study could also highlight the importance of gait impairments and enhance our knowledge of gait assessment protocols as a simple screen to predict major adverse outcomes.

## 2. Methods

### 2.1. Identifying the Research Question

This review primarily aimed to synthesize all of the published evidence on associations between gait impairment and inflammatory markers in older adults. The review question was formulated using the PCC strategy (population, concept, and context): Population: older adults; Concept: Peripheral inflammation and gait impairment; Context: Emerging proinflammatory markers as a powerful tool for identifying high-risk people with gait impairments. The review questions were therefore:Are proinflammatory markers a positive predictor and risk factor for gait impairment?Which proinflammatory markers represent a risk factor for slow gait speed and its complications?

### 2.2. Literature Search Methodology

The following systematic review was performed in line with the Preferred Report Items for Systematic Review and Meta-Analyses (PRISMA). A protocol was structured a priori and designed according to the International Perspective Register of Systemic Review (PROSPERO: CRD42023451108). The research question was developed using the PICO framework. The study analyzed every original article published up to July 2023 that met the following inclusion criteria: (1) full text in English; (2) primary articles only; and (3) presentation of identifiable data measuring gait and inflammatory markers. Studies were searched in the following databases: Excerpta Medica Database (EMBASE), SciVerse (ScienceDirect), Scopus, PubMed, Medline, Web of Science, and the Cochrane Library. The search strategy was formulated based on a combination of controlled descriptors and/or keywords related to the topic and, in addition, a manual search was conducted of the reference lists from the initially selected studies to identify other eligible studies.

An electronic search in PubMed was performed on 25 July 2023 using the following search terms: (interleukin 6[MeSH Terms]) OR (IL-6[MeSH Terms]) AND (gait[MeSH Terms]); (CRP[MeSH Terms]) OR (C reactive protein[MeSH Terms]) OR (protein, C reactive[MeSH Terms]) AND (gait[MeSH Terms]); (tumor necrosis factor[MeSH Terms]) OR (TNFalpha[MeSH Terms]) AND (gait[MeSH Terms]).

When determining which articles to include, we analyzed their title and abstract, then retreived the full text for articles that met the inclusion criteria. Duplicate studies, editorials, conference abstracts, and non-English references were removed using the Rayyan software (https://www.rayyan.ai/) package, and other studies without duplicates were selected based on eligibility criteria by two (L.B. and O.C.-L.) independent and blinded reviewers by reading the abstracts and followed by reading the full text. Another two expert reviewers assessed the internal quality and resolved disagreements in this study selection process (O.C. and F.T.). Finally, the reference lists of all the relevant articles were manually cross-referenced in order to identify any additional articles. These guidelines ensure an adequate evaluation of the research from a methodological point of view in order to exclude possible replication of the methods or results.

Finally, the reference lists of all the relevant articles were manually cross-referenced in order to identify any additional articles. In this study, the following data were extracted: (1) characteristics of the studies (name of the study, authors, year of publication); (2) demographic information of the samples (sample size, participant characteristics of mean age, gender distribution); (3) gait assessment; (4) serum inflammatory parameters (expression level, standard deviation or closest equivalent, correspondent technologies). We excluded conference proceedings, articles reporting results from less than ten elderly patients, that did not assess gait or that assessed gait over a walking distance shorter than five meters

## 3. Results

A number of articles (51) were initially identified while searching the scientific literature regarding this topic, and 21 articles were selected and included in this review. Figure 2 presents a flowchart of the review process (PRISMA diagram).

A qualitative synthesis of the data from the selected studies is provided, describing the inflammatory biomarkers, gait dysfunction, and gait assessment protocol used in each study. Our search selected 21 articles. They are summarized in Table 1. We reported the main findings concerning the relationship between IL-6, CRP, TNFα, and gait parameters. We listed the instruments used and evaluation protocol for gait assessment.

### 3.1. CRP

CRP as an inflammatory marker predictive of decreased gait was the most frequently used study parameter, and was included in 13 of the 21 articles analyzed. High sensitivity CRP (hs-CRP) was used for the relationship with gait parameters in five of them [43,44,45,48,50].

In one study [41], several groups were categorized according to their initial CRP, taking 1.33 mg/dL as a low value, and 2.7 mg/dL as a high value. Values higher than 2.7 mg/dL were associated with a greater probability of subsequent disability. They were also associated with the following: activities of daily living (ADL), instrumental activities of daily living (IADL), impaired balance, impaired walking speed, and arthritis.

Elevated CRP levels were related to low vitamin D levels [42], with a higher probability of slow walking speed, which was attenuated when vitamin D deficiency was absent, although the relevant factor inducing slow gait was CRP. Another study [36] investigated major mobility impairment (MMD) and reported consistent relationships with elevated CRP (>3.0 mg/dL) and a higher risk of MMD, which increased the risk by 38% with a sensitivity of 63% and specificity of 54% as a predictive value over 5 years.

Verghese et al. [43] assessed elevated CRP in a large sample size, with a median of 5.5 mg/dL and minimum levels of 1.1 mg/dL, and found that subjects with elevated values had a reduction in gait speed of −2.46/cm/s/year.

Another study [44] established a relationship between elevated hs-CRP values and the probability of disability in IADL, social and leisure activities (LSA), lower extremity mobility (LEM), and general physical activities (GPA), with knee extensor power, gait speed, or both, explaining the association in continuous variable models with a decrease in these with increased hs-CRP.

Ravaglia et al. [34] used an hs-CRP (ELISA kit with high sensitivity), fibrinogen and leukocyte count as inflammatory markers, and only found an inverse relationship for gait and balance with hs-CRP, and an inverse relationship with CRP and with severe limitation of basic activities of daily living (ABVD) and IADL disability. An increase in hs-CRP levels was negatively associated with gait speed, but this was only significant in older adults, and not in middle-aged men (70).

The relationship between CRP and fibrinogen [49] with gait speed was studied with 373 people over 20 years, associating the two parameters with gait speed as a single task after adjusting for cardiovascular and risk parameters. However, when executive functions were taken into account, the association with CRP disappeared and the association with fibrinogen was attenuated. Dupont et al. [48] used the combination of two parameters, white blood cell count and high-sensitivity CRP, as factors related to the onset of sarcopenia, and observed a negative association with baseline physical activity, quality of life scores, and the SF-36 physical component score in middle-aged and older men. Baseline leukocyte levels were negatively associated with gait velocity, isometric 90° quadriceps and isokinetic peak torque. In another study with 421 participants [46], where CRP was related to motor cognitive risk syndrome (MCR), characterized by cognitive complaints and slow gait, patients with higher CRP levels (2.8 mg/dL) had a higher probability of MCR if memory impairment was present.

### 3.2. Relationship of CRP with Other Biomarkers

Langmann et al. [50] related several inflammatory markers, i.e., hs-CRP, TNFα, and its two receptors (TNFα-R1 and TNFα-R2), interleukin-6 (IL-6), soluble IL-6 receptor (sIL-6R), and interleukin-10 (IL-10) with measures of frailty, function, mobility, and falls in elderly women, and observed that frail patients had significantly higher levels of hs-CRP, TNFα-R1, TNFα-R2, IL-6, and IL-6-sR. Higher baseline levels of hs-CRP and IL-6 were associated with worse physical performance and gait speed at 12 months.

A study of patients with Parkinson’s disease determined hs-CRP levels in patients with freezing of gait (FOG). Patients with FOG showed higher levels of hs-CRP than those without FOG, establishing a FOG-No FOG cut-off level of 0.935 mg/dL, with a sensitivity of 87.1% and specificity of 89.2%.

Penninx [30] evaluated IL-6 together with CRP and the following soluble cytokine receptors (sR), IL-2sR, TNFsR1, TNFsR2, and IL-6sR. People with incident mobility had an IL-6 value of 2.18 mg/L compared to 1.70 mg/L for those without limitations. Soluble cytokine values IL-2sR, TNFsR1, TNFsR2 were also higher. Higher levels of CRP, IL-6, and TNFα were all significantly associated with an increased risk of incident mobility limitation, but appeared to be stronger for IL-6 and CRP than for TNFα. None of the interaction terms for sex or race were statistically significant.

Dupont et al. [51] also analyzed IL-6, IL-8, IL-1β, CRP, and TNFα in sarcopenic subjects. Proinflammatory IL-1β correlated positively with manual grip strength and IL-6 with myofascial autoliberation (aLM). IL-6 correlated inversely with step count. IL-8 was inversely correlated with manual grip strength in women but not in men. In contrast, the proinflammatory cytokines CRP, IL-6, and TNFα correlated inversely with the physical component score of the SF-36 in men, but not in women.

### 3.3. IL-6 and TNF-Alpha

Brown [47] first reported that individuals who had a high IL-6 concentration in blood (3.2 pg/mL) together with previous slow gait and depression, were associated with slow gait. Older individuals in the highest blood IL-6 quartile (more than 4.60 pg/mL) had a 1.75 cm/s/year faster decline in gait speed compared to the participants with IL-6 levels in the lowest quartile [53]. Custodero [37] evaluated the effects of physical performance intervention on 400-m gait speed at 12-month follow-up according to annual IL-6 change categorized by a 1 pg/mL increase or decrease, and observed that subjects with an annual IL-6 change of between −1 and +2 pg/mL had a significant difference in gait speed in the PA intervention group compared with the healthy educational intervention group. In a study of 333 participants with a mean follow-up of 2.3 years, Verghese et al. [7] found that each one-unit increase in log IL-6 levels was associated with a 0.98/cm/s/year increase in the rate of gait velocity decline. Participants in the highest quartile of IL-6 (4.60 pg/mL) had significantly slower gait velocity compared to the other participants. In another study [39], IL-6 levels were positively correlated with knee extensor and knee flexor power. The final model showed that the factors of plasma IL-6 concentrations and physical activity level explained 4.1% of the mean knee flexor power variability, and plasma IL-6 concentration was the variable that best explained the isokinetic variable. Interestingly, participants in the highest tertile of IL-6 (IL-6 > 2.51 pg/mL) were 1.76 times more likely to develop at least mobility-disability and 1.62 times more likely to develop mobility plus ADL-disability, compared with the lowest percentage of IL-6 (1.75 pg/mL) [31].

Nadkarni [38] analyzed IL-6 along with gait speed and white matter (WM) hyperintensities for 10 years in 179 adults. Higher sustained 10-year exposure to IL-6 was significantly associated with slower gait speed. Although attenuated by approximately 47%, this association remained statistically significant after accounting for covariates. A study with 99 participants [52] demonstrated an inverse relationship between circulating IL-6 and thigh muscle composition, strength, and specific weight. Reductions in IL-6 were associated with improvements in gait speed in individuals with higher IL-6 levels (>1.36 pg/mL).

### 3.4. Relationship between IL-6 and TNF-Alpha and Other Biomarkers

The association between IL-6 and major mobility disability (MMD) was investigated [36] and revealed that elevated levels above 2.5 pg/mL were associated with an increased risk of MMD compared to low values ≤2.5 pg/mL. In combination with elevated CRP (>3.0 mg/dLL) alone, patients with both elevated CRP and IL-6 had a 37% increased risk of MMD with a sensitivity of 75% and specificity of 34%.

Kositsawat [40], relating levels of several inflammatory parameters in addition to vitamin D (25 hydroxyvitamin D, 25(OH)D) and insulin-like growthfctor (IGF-1), found that participants with slow walking were more likely to have low 25(OH)D (<20 ng/mL), low IGF-1 (<112 ng/mL), high CRP (≥3 mg/L) and IL- 6 (≥2.87 pg/mL) than participants without a slow walking speed.

## 4. Discussion

Inflammation has adverse affective, cognitive, motor, and neurostructural consequences for older adults [47,54]. In older people, levels of IL-6 higher than 2.5 pg/mL predict a future risk of gait speed decline and a higher risk of functional decline over the subsequent 4 years [7,38]. Elevated sensitive CRP concentrations (≥3 mg/L) at baseline were associated with a faster annual decline in gait velocity of 0.91 cm/s [41] and an increased likelihood of motoric cognitive risk syndrome [46]. In terms of cognitive-motor syndrome, physical function and cognitive function share common neurological processes, and a cognitive decline in tandem with a loss of muscle strength places elderly people at increased risk of personal injury, poor mobility, fall-related injuries leading to frailty, reduced independence, and poorer quality of life [53,55,56,57,58]. In fact, older adults show a high prevalence of gait disorders, and higher IL-6 is linked to a larger volume of white matter hyperintensities (in MRI) that are correlated to slow gait [38,59,60,61,62,63]. Inflammation is common both in aging and frailty, while chronic inflammation is associated with decreased muscle mass and strength, disability, dementia, increased morbidity, and mortality. Studies on the impact of exercise on proinflammatory and anti-inflammatory cytokines have shown that physical exercise in pre-frail older adults in primary care improved depression, gait speed, muscle mass indices, physical function, frailty, and had significant improvement of TNFα levels at 3 months [64,65]. An increase in hs-CRP levels was negatively associated with gait speed only in older adults, but not in middle-aged men [48]. It seems reasonable that over time, chronic inflammation might be a factor affecting denervation of muscles and changes in the neuromuscular junction [18,19,66]. Inflammation also plays a role in the initiation and progression of knee osteoarthritis, and several studies have shown that osteoarthritis is associated with high serum levels of inflammatory markers that alter gait mechanisms [20,21,22,23,24,25].

Biomechanical investigation permits the identification of gait abnormalities that may also adversely affect the quality of life. One of the most important spatiotemporal parameters that seems to significantly decline in patients with chronic inflammation is gait speed. Gait speed is emerging as one of the major determinants of health, but in this review, we did not find one particular gait assessment protocol or tool that is used by most of the studies, but instead a wide variety of combinations are described. Physical performance and gait speed was assessed from 4 m to 400 m on an instrumented walkway [29,36,38,52] making comparison of the results difficult. We excluded gait tests with a walking distance of less than 4 m in order to only include tests affected by the start and the stop phases of gait to a lesser extent.

The large number of patients analyzed in the studies included (an average of 1300 patients, with a minimum of 40 and a maximum of 5642) suggests that the peripheral inflammatory biomarkers IL-6 and CRP are related to gait impairments. A consensus among the clinical research community in this regard has yet to be achieved in order to standardize the evidence according to which higher levels of CRP, IL-6, and TNFα are all significantly associated with a faster decline in gait speed, worse physical performance, major mobility impairment, lower extremity mobility, impairments to activities of daily living, and general physical activities.

Gait speed is the most frequently recorded parameter in the literature, and 0.8 m/s is a reliable cut-off for identifying subjects at increased risk of disability but further studies should be conducted to select the most sensitive protocol for assessing and representing this biomarker of frailty in people with inflammation. Many circulating biomarkers change with age independently of disease [67,68]. Characterizing age-adjusted distributions of these biomarkers in healthy older adults would therefore be important to increase specificity of diagnosis, inform treatment and prevention, and limit unnecessary procedures and treatments. Despite the fact that people over the age of 65 are the fastest growing age segment of our population, characterization of many circulating biomarkers for the different age groups and gender remain incomplete, and changes at extreme old ages are predicted using mathematical models [69]. In this review, we only identified analysis of age groups in the study of Dupont and co-workers (2021) who performed a stratified cross-sectional analysis according to two age groups, 40–59 years (middle-aged adults) and 60–79 years (older adults), demonstrating that hs-CRP concentration in blood was significantly inversely associated with SF-36 physical component scores in both middle-aged and older adults. This aspect needs to be clearly investigated in future studies in this field.

A limitation of any systematic review is the potential omission of relevant articles. Although we tried to use exhaustive inclusion criteria, it is possible that we did not identify all publications on the subject. Our search strategy was based on MeSH and key words assigned by the authors, and we may have missed publications that were not indexed under those terms, although we tried to identify further articles through reference lists. However, the search strategy used in this work had the advantage of using five large databases, enabling an exhaustive literature review in commonly used biomedical databases for literature search retrieval.

## 5. Conclusions

Substantial evidence suggests that the peripheral inflammatory biomarkers IL-6 and CRP are related to frailty status and gait impairment. Cytokines are key chemical mediators of the immune response produced by cells, and an understanding of the molecular, biomechanical, and neuropsychological networks involved is required in order to enhance prevention strategies of gait impairments and its complications, such as falls among the aging population. Our findings regarding the causal relationship between chronic inflammation and gait impairment are not conclusive due to heterogeneous approaches to gait analysis in terms of the instruments used and measurement duration/length. Walking speed is a commonly used outcome in different types of studies, but the methodologies and descriptions of walking tests vary widely from study to study. We recommend standardizing the protocol for assessing walking speed in older adults, such as using a 6 or 10-Meter Walk Test or a 6-min walk test in order to understand what could also benefit from standardization [70,71,72,73].

## Figures and Tables

**Figure 1 ijms-25-01368-f001:**
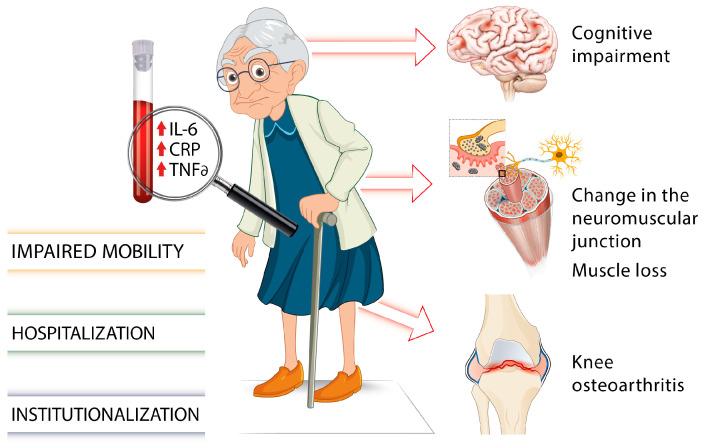
Involvement of proinflammatory markers on gait impairments and frailty in older individuals [7,18,19,20,21,22,23,24,25,26,27,28,29,30,31,32,33,34,35].

**Figure 2 ijms-25-01368-f002:**
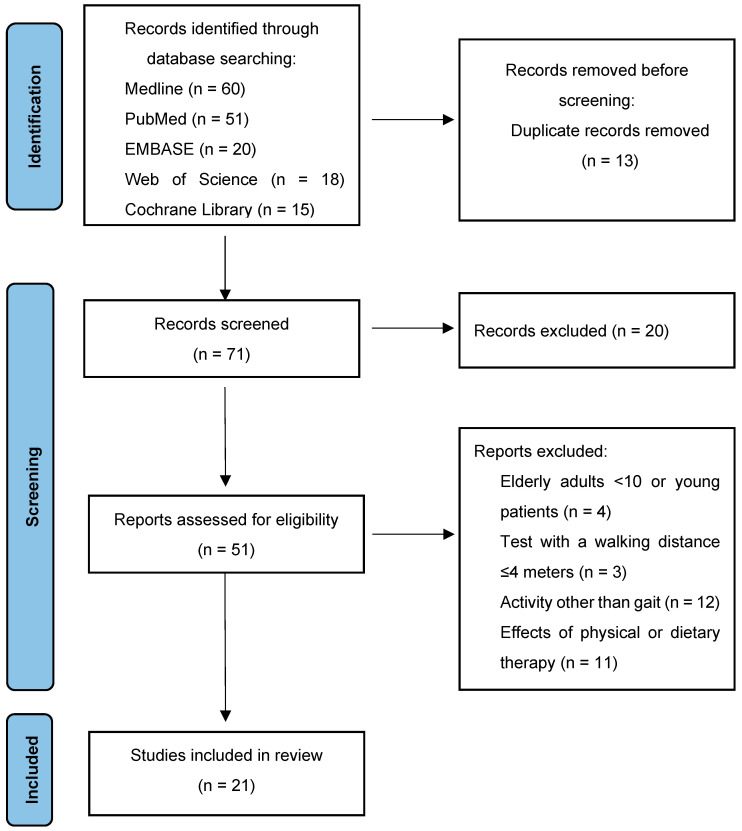
PRISMA diagram: the figure represents the study selection flow through identification, screening, eligibility and inclusion.

**Table 1 ijms-25-01368-t001:** Characteristics of inflammatory biomarkers and gait parameters: an overview of the selected studies.

Reference	Sample Characteristics	Inflammatory Biomarker	Gait Assessment	Design of the Study	Main Findings
Verghese et al., 2012 [7]	333 participants. Age: 80.27 (5.48)	IL-6 and TNFα	GAITRite^®^. Participants walked for two trials (2–3 h apart) on a walkway 15 feet long	Cross-sectional and longitudinal analyses	IL-6 at baseline predicted future risk of gait speed decline; each one-unit increase in log IL-6 (which corresponds to an increase of 1.7 times in untransformed IL-6 levels)
Penninx et al., 2004 [30]	2979 participants. Age: 73.5 ± 2.9	IL-6, TNFα, CRP, IL-6R, IL-2R, TNFsR1, and TNFsR2	Difficulty or inability to walk a quarter of a mile or climb 10 steps in two consecutive six-month follow-up assessments.	Prospective cohort study	Persons who developed incident mobility limitation had significantly higher serum levels of IL-6 (2.18 pg/mL), TNFα (3.36 pg/mL), and CRP (2.31 mg/L) at baseline.
Ferrucci et al., 1999 [31]	350 participants. Age: 79.4 ± 0.3	IL-6	Difficulty in mobility or ADL (walking half a mile or climbing a flight of stairs, walking across a small room, bathing, transferring from bed to chair, and using the toilet).	Longitudinal case-based design.	Older persons who are completely independent in ADL and mobility and have circulating levels of IL-6 greater than 2.5 pg/mL are at higher risk of functional decline over the subsequent 4 years
Beavers et al., 2021 [36]	1732 participants. Age: 77.5 ± 5.1	IL-6 and CRP	Walking 400 m in less than 15 min.	Observational pooled analysis	High baseline IL-6 and CRP were associated with an increased risk of major mobility disability among older adults with slow gait speed (<1.0 m/s)
Custodero et al., 2023 [37]	1300 participants. Age: 79.14 ± 5.27	IL-6	400 m gait speed	Multicenter single-blind randomized clinical trial	No effects were observed on 400 m gait speed for wider range of variation of plasma IL-6 levels
Nadkarni et al., 2016 [38]	179 participants. Age: 83.1 (2.7)	IL-6	Gait speed was measured on a 4-m-long GaitMatII (EQ Inc., Chalfont, PA, USA).	Prospective cohort study	Sustained exposure to high IL-6 over 10 years rather than the rate of change in IL-6 or an isolated high IL-6 level may adversely affect gait speed
Carvalho Felicio et al., 2014 [39]	221 participants. Age: 71.07 ± 4.93	IL-6 and sTNFR1	Q&Q stopwatch (CBM Corp., Japan, Tokyo) was used to measure habitual and fast gait velocity over 10 m.	Cross-sectional study	There was no negative correlation between inflammatory mediators and muscle or physical performance in elderly women.
Kositsawat et al., 2020 [40]	Participants: 713 at 3-year visit and 600 at 6-year visit. Age: 74.6 (7.1)	IL-6, CRP and IGF-1	Gait speed was assessed over a 4-m walk at every visit.	Prospective observational cohort study	Participants with slow gait speed (<0.8 m/s) had high CRP (≥3 mg/L) and high IL-6 (≥2.87 pg/mL)
Lassale et al., 2019 [41]	2437 participants aged 47–87 years	hs-CRP	Gait speed was measured directly, with respondents aged 60 years and older walking a distance of 8 feet (2.4 m) twice: the mean speed (m/s) of the two trials was used.	Prospective cohort study	Stable-high trajectory (5.7 mg/L) was associated with low grip strength, walking speed, and balance impairment, but the precision of the estimates was low and they did not reach statistical significance at the conventional level
Kositsawat et al., 2013 [42]	1826 participants aged 50–85 years	CRP and Vitamin D	Participants were asked to walk 20 feet (6.1 m) unassisted or with or without a cane or a walker at usual speed. Slow gait speed was defined as <0.8 m/s.	Cross-sectional study	Participants with high CRP without severe vitamin D deficiency were more likely to have slow gait speed.
Verghese et al., 2012 [43]	624 participants. Age: 80.34 (5.37)	hs-CRP	Subjects walked on an instrumented walkway (GAITRite^®^, CIR systems, Havertown, PA, USA).	Prospective cohort	Elevated hs-CRP levels (≥3 mg/L) at baseline were associated with a faster annual decline in gait velocity of 0.91 cm/s
Kuo et al., 2006 [44]	1680 participants. Age >60 years old	hs-CRP	20-foot timed walk test.	Cross-sectional study	CRP had an inverse relationship to leg power and walking speed
Liu et al., 2022 [45]	217 participants. Aged: 68.83 ± 9.50	CRP	Freezing of gait was diagnosed in Parkinson’s disease patients who scored at least 1 on the third Freezing of Gait score.	Prospective study	A cut-off level of 0.935 mg/L distinguished patients with or without Freezing of gait (FOG). hs-CRP levels in plasma can be used as a sensitive biomarker of FOG and of Parkinson’s disease progression
Bai et al., 2021 [46]	5642 participants. Age: 68.13 ± 6.55	CRP	The average time respondents took to walk along a straight path twice was used to compute gait speed.	Longitudinal Study	Participants with higher CRP levels had an increased likelihood of motoric cognitive risk syndrome
Brown et al., 2016 [47]	3075 participants. Age: 73.6 (2.87)	IL-6	Participants’ usual walking speed was assessed over 6, 10, and 20 m.	Longitudinal Study	Slow gait was associated with inflammation and depression
Ravaglia et al., 2004 [34]	739 participants. Age: 74.0 ± 6.6	hs-CRP, fibrinogen, leucocyte count, cholesterol, and albumin	Participants underwent the Tinetti test for gait and balance.	Cross-sectional study	CRP (>0.4 mg/dL) was consistently associated with a Tinetti test score below 19
Dupont et al., 2021 [48]	2577 participants. Aged: 59.66 ± 11.00	hs-CRP, WBC, and Albumin	Physical performance was assessed by gait speed.	Cross-sectional study	An increase in hs-CRP levels was negatively associated with gait speed; only significant in older adults, but not in middle-aged men.
Heumann et al., 2022 [49]	373 participants. Aged: 64 ± 13	CRP, Fibrinogen	Participants were asked to walk for 1 min or ten meters as a Single Task (ST) and a Dual Task (DT).	Population-based longitudinal study	Gait speed under ST was negatively associated with baseline levels of CRP
Langmann et al., 2017 [50]	178 participants	IL-6, TNFα, hs-CRP	Gait speed was assessed with the Fried frailty index on a 6-m walk	Prospective study	Higher baseline hs-CRP and IL-6 levels were associated with worse ADL performance, physical performance, and gait speed at 12 months
Dupont et al., 2023 [51]	40 participants. Aged: 77.1 (6.8)	IL-6, TNFα, CRP, IL-1β, IL-8	The subject was instructed to walk six meters at a usual pace.	Cross-sectional study	CRP, IL-6, and TNFα have an inverse correlation with the SF-36 physical component score. This study highlights an important role of gender
Grosicki et al., 2020 [52]	95 participants. Aged: 78 (5)	IL-6	Walking speed over 400 m (m)	Randomized controlled trial	Reductions in IL-6 were associated with gait speed improvements in “higher” IL-6 individuals (>1.36 pg/mL)

## Data Availability

Data supporting reported results can be found writing at lorenzo.brognara2@unibo.it.

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
