# Peer review of "Inflammatory Biomarkers and Gait Impairment in Older Adults: A Systematic Review"

_ijms, 2024, doi:10.3390/ijms25031368_

Round 1

Reviewer 1 Report

Comments and Suggestions for Authors

Major revision

Comments on the Quality of English Language

need major revision , seems to be very less writeup for an review paper

Author Response

The title, "Inflammatory Biomarkers and Gait Impairment in Older Adults: A Systematic Review," immediately captures the essence of the study. It provides a clear focus on the relationship between inflammatory biomarkers and gait impairment specifically in the older adult population. The use of the term "systematic review" indicates a rigorous and comprehensive approach to synthesizing existing literature on the subject. The inclusion of the term "inflammatory biomarkers" sets the stage for an exploration into the molecular aspects that may contribute to gait impairments in older individuals. This choice of focus suggests an in-depth investigation into the potential connections between peripheral inflammation and alterations in gait speed, providing a hint at the clinical relevance and implications of the study.

The authors would like to thank the reviewer for recognizing the importance of the topic of our research and for kind consideration.

Add some more points regarding systematic approach employed in identifying relevant peer- reviewed articles. The use of five electronic databases, including Medline, PsycINFO, CINAHL, EMBASE, and PubMed, indicates   a comprehensive literature search strategy. The inclusion criteria are clearly defined, specifying the focus on older adults, spatiotemporal gait characteristics, proinflammatory markers, and the language of the articles (English). This transparency in methodology enhances the credibility of the study.

-Thanks for highlighting, we detailed the systematic approach employed in identifying relevant peer- reviewed articles as suggested:

  1. 103 - 113

2.1. Identifying the research question

This review primarily aimed to synthesise all published evidence on associations between gait impairment and chronic inflammation in elderly adults. The review question was formulated using the PCC strategy (population, concept and context). Population: older adults; Concept: Peripheral inflammation and gait impairment; Context: Emerging proinflammatory markers as a powerful tool for identifying high-risk people with gait impairments. Thus, the review question was:

  • Are proinflammatory markers a positive predictor and risk factor for gait impairment?
  • Which proinflammatory markers represent a risk factor for the slow gait speed and its complications?

  1. 120-128

The study analyze every original article in the PubMed/Medline and Scopus electronic bibliographic databases, published until july 2023, which met the following inclusion criteria: (1) full text in English;

(2) primary articles only; and (3) presentation of identifiable data measuring gait and inflammatory markers. Studies were searched in the following databases: Excerpta Medica Database (EMBASE), SciVerse (ScienceDirect), Scopus, PubMed, Medline, Web of Science, Cochrane Library. The search strategy was formulated from a combination of controlled descriptors and/or keywords related to the topic and, in addition, a manual search was conducted on the reference lists from the initially selected studies to identify other eligible studies.

  1. 134-144

When determining which articles to include, we analyzed their title and abstract, and the full text was then retrieved for articles that met the inclusion criteria. Duplicates studies editorial conference abstracts and non-English references were removed used Rayyan software and other studies, without duplicates, were selected based on eligibility criteria by two (L.B and O.C-L) independent and blinded reviewers by reading the abstracts and followed by reading the full text. Other two expert reviewer assessed internal quality and solved disagreements in this study selection process (O.C and F.T). Finally, the reference lists of all the relevant articles were manually cross-referenced in order to identify any additional articles. These guidelines ensure an adequate evaluation of the research from a methodological point of view in order to exclude possible replication of the methods or results.

L.191-194

  1. Results

A qualitative synthesis of the selected studies’ data are provided, describing the inflammatory biomarkers, gait dysfunction and gait assessment protocol used in each study. A descriptive table summarizes all this information.

  • Include at least 1 more figure and table to enrich the manuscript 90

-      Done: Figure 1. Involvement of proinflammatory markers on gait impairments and frailty in older individuals [18-37]

The decision not to perform a meta-analysis due to the heterogeneity of the studies is justified, and the alternative approach of narratively synthesizing the results is explained. This transparency in reporting methodological decisions adds depth to the abstract, allowing the reader to understand the challenges faced and the rationale behind the chosen methodology.

-Thank you very much for these comments.

Include more recent references part, highlighting the role of specific inflammatory biomarkers, namely TNF-α, CRP, and IL-6, in gait impairment. The mention of IL-

6 as an effective biomarker in diagnosing slow gait speed adds a practical dimension to the findings. The call for further longitudinal research to establish the use of molecular biomarkers in monitoring gait impairment sets the stage for futureinvestigations and applications in clinical practice.

We deeply appreciate the reviewer’s suggestion. According to the reviewer’s comment, we have

added more recent references highlighting the role of inflammatory biomarkers in gait impairment.

Discussion:

  1. 14-20

“Inflammation is common both in aging process and frailty, and it is associated with decreased muscle mass and strength, disability, dementia, increased morbidity, and mortality. Studies on impact of exercise on pro-inflammatory and anti-inflammatory cytokines have shown that physical in pre- frail older adults in primary care improved depression, gait speed, muscle mass indices, physical function, frailty and had significant improvement of TNF-α levels at 3 months [68-69].”

In summary, the effectively communicates the scope, methodology, and key findings of the systematic review, offering a comprehensive overview of the study's contribution to understanding the link between inflammatory biomarkers and gait impairment in older adults.

-Thank you very much for these comments.

Reviewer 2 Report

Comments and Suggestions for Authors

The manuscript provides a good systematic review on the association between inflammatory biomarkers (especially IL-6, CRP, and TNF-alpha) and gait impairment in older adults. The rationale, objectives, methods, results, and conclusions are clearly presented.

Specific Suggestions

  • Standardize abbreviations for consistency (e.g. sometimes write out interleukin-6 vs. IL-6)
  • Add page numbers
  • Proofread minor typos (biomaker, hetoregeneous)
  • Consider additional limitations - did the review account for differences in age groups or health status among studies?

Additional comments:

- The main question addressed is whether proinflammatory markers are a positive predictor for gait impairments and their complications, such as falls in older adults.
- I consider the topic to be original and relevant. It addresses the specific gap in understanding the association between inflammatory biomarkers and gait impairment.
- The review synthesizes evidence on the links between specific inflammatory markers (IL-6, CRP, TNF-alpha) and measures of gait and physical impairment. It highlights potential biomarker roles for diagnosis and monitoring.
- Regarding methodology, the authors could consider additional criteria for study inclusion, such as controlling for comorbidities that affect gait. Additional longitudinal studies would allow stronger conclusions about causality.
- The conclusions are consistent with the evidence presented. The authors highlight the potential role for IL-6 and CRP as biomarkers related to gait declines, but note that further research is needed.
- The references seem appropriate and directly support the statements made. A wide range of relevant studies were cited.
- The manuscript has one figure and one table. This is fine for a review paper.

Comments on the Quality of English Language

Proofread minor typos (biomaker, hetoregeneous)

Author Response

The manuscript provides a good systematic review on the association between inflammatory biomarkers (especially IL-6, CRP, and TNF-alpha) and gait impairment in older adults. The rationale, objectives, methods, results, and conclusions are clearly presented.

-We appreciate your positive evaluation of our work. Specific Suggestions

  • Standardize abbreviations for consistency (e.g. sometimes write out interleukin-6 IL-6)

-Done as suggested by the Reviewer.

  • Add page numbers

-Done as suggested (in the top of each page).

  • Proofread minor typos (biomaker, hetoregeneous)

-Thanks for highlighting, now , the revised manuscript was proofread by an English editing service .

  • Consider additional limitations - did the review account for differences in age groups or health status among studies?
    • Thanks for this interesting suggestion. We have added the lack of analysis of age groups in the revised version of the Discussion section as follows:

L.48-60

“Many circulating biomarkers change with age independently of disease (Kubota et al., 2012; Sebastiani et al., 2016). Characterizing age-adjusted distributions of these bi-omarkers in healthy older adults would therefore be important to increase specificity of diagnosis, inform treatment and prevention, and limit unnecessary procedures and treatments. Despite the fact that people over the age of 65 are the fastest growing age seg-ment of our population, characterization of many circulating biomarkers for the different age groups and gender remain incomplete, and changes at extreme old ages are predicted from mathematical models (Arbeev et al., 2011). In this review, we only identified analysis of age groups in the study of Dupont and co-workers (2021) that performed a stratified cross-sectional analysis according to age groups 40–59 years (middle-aged adults) and 60–79 years (older adults) demonstrating that hs-CRP concentration in blood was signifi-cantly inversely associated with SF-36 physical component scores in both middle-aged and older adults. This aspect needs to be clearly investigated in future studies in this field.”

  • Kubota K, Kadomura T, Ohta K, Koyama K, Okuda H, Kobayashi M, et al. Analyses of laboratory data and establishment of reference values and intervals for healthy elderly J Nutr Health Aging. 2012;16(4):412–416.
  • Sebastiani P, Thyagarajan B, Sun F, Honig LS, Schupf N, Cosentino S, Feitosa MF, Wojczynski M, Newman AB, Montano M, Perls TT. Age and Sex Distributions of Age-Related Biomarker Values in Healthy Older Adults from the Long Life Family J Am Geriatr Soc. 2016 Nov;64(11):e189-e194.
  • Arbeev KG, Ukraintseva SV, Akushevich I, Kulminski AM, Arbeeva LS, Akushevich L, et al. Age trajectories of physiological indices in relation to healthy life Mech Ageing Dev. 2011;132(3):93–102.

Additional comments:

  • The main question addressed is whether proinflammatory markers are a positive predictor for gait impairments and their complications, such as falls in older adults.

I consider the topic to be original and relevant. It addresses the specific gap in understanding the association between inflammatory biomarkers and gait impairment.

-We appreciate the reviewer’s positive evaluation of our work.

  • The review synthesizes evidence on the links between specific inflammatory markers (IL-6, CRP, TNF- alpha) and measures of gait and physical It highlights potential biomarker roles for diagnosis

and monitoring. Regarding methodology, the authors could consider additional criteria for study inclusion, such as controlling for comorbidities that affect gait.

-Thank you for this interesting suggestion, but due to limited number of studies (N=21) it was not possible to summarize the results controlling for different comorbidities, also because large sample’ studies such as the studies by Penninx et al. 2004, Beavers et al. 2021, Custodero et al. 2023, Lassale et al. 2019 , Bai et al. 2021, Brown et al. 2016 , and Dupont et al. 2021, all were performed in community-dwelling older adults which were selected without a clearly defined comorbidity’ inclusion criteria.

  • The conclusions are consistent with the evidence presented. The authors highlight the potential role for IL-6 and CRP as biomarkers related to gait declines, but note that further research is

The references seem appropriate and directly support the statements made. A wide range of relevant studies were cited.

-Thank you.

The manuscript has one figure and one table. This is fine for a review paper.

-The authors would like to thank the reviewer for recognizing the importance of the topic of our research and for kind consideration. Our deepest gratitude goes to you for your careful work and thoughtful suggestions that have helped improve this paper substantially

Reviewer 3 Report

Comments and Suggestions for Authors

This article presents an interesting topic. Please see below points for major revision:

Why were only the mentioned databases used by the authors in their search?

The title should be changed into narrative review (not systematic review, if this is the real type of the paper).

The hypotheses should be clearly described (end of the Introduction).

2.1. looks not supported by relevant literature. Please add in order to support.

Please explain in more detail to the reader: When determining which articles to include, we analyzed their title and abstract, and the full text was then retrieved for articles that met the inclusion criteria.

In the Conclusion about the 10-meter walk test, authors can also add the 6-minute walk test and its importance as mentioned in relevant assesment https://doi.org/10.1080/23279095.2020.1870465

Additional comments:

The main question of this review refers is: Proinflammatory markers are a positive predictor for gait impairments and their complications, such as falls in older adults? This is the main question in the Abstract, but in the Methods section more questions are reported and they should be supported by relevant literature.

The topic refers to a gap in the field and therefore it is original. This research adds a more systematic approach to already published studies.

The databases should be more (some only are mentioned that were searched). In addition to that, the methodology should be described in more detail.

A more detailed description is needed regarding the included studies.

The references are appropriate for the majority of them and The tables are ok.

Comments on the Quality of English Language

Moderate English language editing is necessary throughout the text.

Author Response

This article presents an interesting topic.

-Thank you.

Please see below points for major revision:

Why were only the mentioned databases used by the authors in their search?

Thanks for highlighting, we have selected the seven well-known biomedical databases such as Excerpta Medica Database (EMBASE), SciVerse (ScienceDirect), Scopus, PubMed, Medline, Web of Science, Cochrane Library mostly used in biomedical sciences using english as international language for scientific purposes. We have detailed as suggested :

L.16-25

“The systematic review was performed in line with the Preferred Report Items for Systematic Review and Meta-Analyses (PRISMA). A protocol for literature searches was structured a priori and designed according to the International Perspective Register of Systemic Review (PROSPERO: CRD42023451108).. Peer-reviewed original articles were identified by searching seven electronic databases: Excerpta Medica Database (EMBASE), SciVerse (ScienceDirect), Scopus, PubMed, Med-line, Web of Science, Cochrane Library. The search strategy was formulated from a combination of controlled descriptors and/or keywords related to the topic and, in addition, a manual search was conducted on the reference lists from the initially selected studies to identify other eligible studies.“

We have added the following limitation at th end of the Discussion section of teh revised version of the manuscript:

Discussion chapter: L.61-68

“A limitation of any systematic review is the potential omission of relevant articles. Alt-hough we tried to use exhaustive inclusion criteria, it is possible that we did not identify all publications on the subject. Our search strategy was based on MeSH and key words assigned by authors, and we may have missed publications that were not indexed under these terms, although we tried to identify further articles through reference lists. However, the search strategy used in this work had the advantage of using five large databases, en-abling an exhaustive literature review in commonly used biomedical databases for litera-ture search retrieval.”

The title should be changed into narrative review (not systematic review, if this is the real type of the paper).

We prefer to keep systematic review as the review was previously registered as systematci review in PROSPERO database with reference number CRD42023451108 (see atatched pdf file from PROSPERO REGISRED SISTEMATIC REVIEW).

The hypotheses should be clearly described (end of the Introduction).

Thank you very much for pointing it out. We have clarified this aspect in the introduction section.

L.93-101

“the aim of this systematic review is to analyze current scientific knowledge and the hypothesis which is why proinflammatory markers represent a positive predictor for gait impairment and its complications. According to the results acquired from this study, there’ll be more opportunities for developing optimistic hypothesis insights correlation between proinflammatory markers as a powerful tool for identifying high-risk people with gait impairments, as well as, the results of this study could also highlight the importance of gait impairments

and to promote our knowledge about gait assessment protocol as a simple screen to predicts major adverse outcomes.”

  • looks not supported by relevant Please add in order to support.

Thanks for highlighting, done as suggested

  1. 103-113

“2.1. Identifying the research question

This review primarily aimed to synthesise all published evidence on associations between gait impairment and chronic inflammation in elderly adults. The review question was formulated using the PCC strategy (population, concept and context). Population: older adults; Concept: Peripheral inflammation and gait

impairment; Context: Emerging proinflammatory markers as a powerful tool for identifying high-risk people with gait impairments. Thus, the review question was:

  • Are proinflammatory markers a positive predictor and risk factor for gait impairment?
  • Which proinflammatory markers represent a risk factor for the slow gait speed and its complications?

Please explain in more detail to the reader: When determining which articles to include, we analyzed their title and abstract, and the full text was then retrieved for articles that met the inclusion criteria.

Done as suggested

L.134-144 “When determining which articles to include, we analyzed their title and abstract, and the full text was then retrieved for articles that met the inclusion criteria. Duplicates studies editorial conference abstracts and non-English references were removed used Rayyan software and other studies, without duplicates, were selected based on eligibility criteria by two (L.B and O.C-L) independent and blinded reviewers by reading the abstracts and followed by reading the full text. Other two expert reviewer assessed internal quality and solved disagreements in this study selection process (O.C and F.T). Finally, the reference lists of all the relevant articles were manually cross-referenced in order to identify any additional articles. These guidelines ensure an adequate evaluation of the research from a methodological point of view in order to exlude possible replication of the methods or results.”

In the Conclusion about the 10-meter walk test, authors can also add the 6-minute walk test and its importance as mentioned in relevant assesment https://doi.org/10.1080/23279095.2020.1870465

We added as suggested Additional comments:

The main question of this review refers is: Proinflammatory markers are a positive predictor for gait impairments and their complications, such as falls in older adults? This is the main question in the Abstract, but in the Methods section more questions are reported and they should be supported by relevant literature.

According to Reviewer’s suggestion we have specified in the abstract the aimes according to Methods

section as follows:

L.13-16“The aim of this systematic literary review is to determine whether proinflammatory markers are a positive predictor for gait impairments and their complications, such as falls in older adults and may represent a risk factor for the slow gait speed and its com-plications.”

The topic refers to a gap in the field and therefore it is original. This research adds a more systematic approach to already published studies.

The authors would like to thank the reviewer for recognizing the importance of the topic of our research and for kind consideration

The databases should be more (some only are mentioned that were searched). In addition to that, the methodology should be described in more detail.

Done as suggested,

A more detailed description is needed regarding the included studies. Done as suggested.

The references are appropriate for the majority of them and The tables are ok.

We have consider all suggestions and either make appropriate changes in the manuscript providing suitable replies. We hope to have improved the paper. Thank you for your encouraging review of our manuscript. In particular, we wanted to thank you for your constructive suggestions.

Round 2

Reviewer 1 Report

Comments and Suggestions for Authors

all corrections are carried out
the paper might be accepted

Comments on the Quality of English Language

All the corrections are carried out, paper may be accepted